# Identification of Cyanobacterial Strains with Potential for the Treatment of Obesity-Related Co-Morbidities by Bioactivity, Toxicity Evaluation and Metabolite Profiling

**DOI:** 10.3390/md17050280

**Published:** 2019-05-10

**Authors:** Margarida Costa, Filipa Rosa, Tiago Ribeiro, Rene Hernandez-Bautista, Marco Bonaldo, Natália Gonçalves Silva, Finnur Eiríksson, Margrét Thorsteinsdóttir, Siegfried Ussar, Ralph Urbatzka

**Affiliations:** 1Faculty of Pharmaceutical Sciences, University of Iceland, Hofsvallagata 53, 107 Reykjavik, Iceland; costa.anamarg@gmail.com (M.C.); margreth@hi.is (M.T.); 2Interdisciplinary Centre of Marine and Environmental Research (CIIMAR/CIMAR), University of Porto, Avenida General Norton de Matos, s/n, 4450-208 Matosinhos, Portugal; frosa@ciimar.up.pt (F.R.); tribeiro@ciimar.up.pt (T.R.); nsilva@ciimar.up.pt (N.G.S.); 3RG Adipocyte and Metabolism, Institute for Diabetes and Obesity, Helmholtz Center Munich, 85764 Neuherberg, Germany; rene.hernandez@helmholtz-muenchen.de (R.H.-B.); siegfried.ussar@helmholtz-muenchen.de (S.U.); 4INBB, Consorzio Interuniversitario Biosistemi e Biostrutture, 00136 Rome, Italy; marcobonaldo90@libero.it; 5ArcticMass, Sturlugata 8, 101 Reykjavik, Iceland; finnur@arcticmass.is

**Keywords:** anti-obesity drugs, metabolite profiling, zebrafish Nile red fat metabolism assay, uncoupling protein 1, bioactivity screening, diabetes, fatty liver disease, cyanobacteria

## Abstract

Obesity is a complex disease resulting in several metabolic co-morbidities and is increasing at epidemic rates. The marine environment is an interesting resource of novel compounds and in particular cyanobacteria are well known for their capacity to produce novel secondary metabolites. In this work, we explored the potential of cyanobacteria for the production of compounds with relevant activities towards metabolic diseases using a blend of target-based, phenotypic and zebrafish assays as whole small animal models. A total of 46 cyanobacterial strains were grown and biomass fractionated, yielding in total 263 fractions. Bioactivities related to metabolic function were tested in different *in vitro* and *in vivo* models. Studying adipogenic and thermogenic gene expression in brown adipocytes, lipid metabolism and glucose uptake in hepatocytes, as well as lipid metabolism in zebrafish larvae, we identified 66 (25%) active fractions. This together with metabolite profiling and the evaluation of toxicity allowed the identification of 18 (7%) fractions with promising bioactivity towards different aspects of metabolic disease. Among those, we identified several known compounds, such as eryloside T, leptosin F, pheophorbide A, phaeophytin A, chlorophyll A, present as minor peaks. Those compounds were previously not described to have bioactivities in metabolic regulation, and both known or unknown compounds could be responsible for such effects. In summary, we find that cyanobacteria hold a huge repertoire of molecules with specific bioactivities towards metabolic diseases, which needs to be explored in the future.

## 1. Introduction

The worldwide prevalence of obesity as a modern and imminent health hazard is clear and very well documented [1]. If the current growth rates are maintained, 38% of the global adult population will be overweight and 28% obese by 2030 [2]. Obesity is defined by a body mass index (BMI) greater than 30, and associated with complex co-morbidities such as type 2 diabetes, cardiovascular disease and several types of cancer [3,4,5,6]. Exercise and healthy nutrition show only limited effects on weight loss and patients tend to get back to or exceed the original weight after only a few years. Furthermore, many anti-obesity drugs have adverse side-effects. One example is Sibutramine, which was withdrawn in 2010 due to cardiovascular events and strokes [7]. In contrast, bariatric surgery is by far the most effective treatment for obesity, however it has significant risk for complications and only a fraction of obese patients is eligible for these operations. 

Therefore, new sources of anti-obesity drugs and therapies are urgently needed. A current strategy seems to return to basic natural product drug discovery [8]. Natural products are mostly secondary metabolites from macro- and microorganisms that evolved through time to target specific molecules. The potential of most natural products is still underexplored, in particular in marine environments, and may represent a promising source of new anti-obesity agents, as already reported in a few studies using marine cyanobacteria or marine sponge-associated fungi [9,10]. 

Cyanobacteria are a group of gram-negative prokaryotes widespread in the planet with numerous biosynthetic routes that lead to structural diverse and biologically active secondary metabolites [11]. The Interdisciplinary Centre of Marine and Environmental Research (CIIMAR) hosts a cyanobacterial culture collection (LEGE-cc) with approx. 400 strains mainly collected in freshwater, estuarine and marine environments [12]. A small part of this chemical diversity was already explored for the identification of anti-cancer activities [13]; however, the anti-obesity potential has not been analyzed before. 

The aim of this work was to identify cyanobacterial strains with the potential to produce promising secondary metabolites that have strong bioactivities towards obesity or obesity-related co-morbidities. The screening focused on obesity and obesity-related bioactivities using cellular models *in vitro* and physiologically relevant whole small animal models *in vivo*. Effects on lipid homeostasis were analyzed in zebrafish, while effects on glucose and lipid metabolism were studied in human HepG2 cells and complemented by analysis of compound activity towards adipocyte differentiation and thermogenic gene expression in murine brown adipocytes. Metabolite profiling and toxicity studies further narrowed the selection to the most promising 18 (7%) fractions. Produced metabolites were categorized into known or unknown compounds by data base searches.

## 2. Results

### 2.1. Lipid Reducing Activity in Zebrafish Larvae

The library of cyanobacterial fractions was screened for lipid reducing activity using the zebrafish Nile red fat metabolism assay. From the 263 analyzed fractions, 17 (6.5%) reduced the mean fluorescence intensity (MFI) >50%, while 29 (11%) diminished the MFI >30% (Figure 1A,B). The most promising fractions belong to 15 different cyanobacterial strains (12, 23, 130, 131, 141, 144, 161, 180, 187, 193, 196, 226, 232, 250, 256, 259, 262), with the majority from marine ecosystems (73%).

### 2.2. Anti-Steatosis Activity in HepG2 Cells

HepG2 cells were fed with 62 µM sodium oleate in order to induce the formation of lipid droplets. Fat overloading works similarly in primary hepatocytes and HepG2 cells and is an established *in vitro* model for hepatic steatosis [14]. Reduction of lipids was quantified after 6 h exposure to oleate and individual cyanobacterial fractions. Oleate exposure increased HepG2 lipid content compared to cells cultured in regular cell culture medium (Figure 1C,D). Among the 263 analyzed fractions, 50 (19%) reduced lipid content (mean fluorescence intensity of Nile red; MFI >30%), and 32 (12.2%) reduced the MFI >50%. Taking into account the toxicity of some fractions, as detailed below, the most promising fractions (58, 77, 89, 90, 101, 102, 107, 108, 109, 177, 178, 192, 199, 202, 220, 231, 232) were derived from 11 cyanobacterial strains and 54.4% belong to marine ecosystems.

### 2.3. Effects on Brown Adipocyte Differentiation and Thermogenic Gene Expression

Uncoupling protein-1 (UCP-1) is a brown adipocyte specific gene, which uncouples the mitochondrial respiratory chain to produce heat instead of ATP. Thus, expression of UCP-1 is an indicator for the thermogenic capacity of brown adipocytes and for distinguishing energy storing white from energy dissipating brown adipocytes. However, gene expression analysis is just an indicator of the functional protein, and hence, future confirmation is necessary. Assessment of UCP-1 gene expression upon treatment with 10 µg mL^−1^ of each individual cyanobacterial fraction during differentiation revealed significantly increased expression after exposure to fractions 168, 228, 229 and 232 (Figure 2A). The cyanobacterial fractions were tested on their effect on brown adipocyte differentiation, as monitored by expression of the key adipogenic transcription factor PPARγ. As shown in Figure 2B, the fractions 139, 141, 142, 155 and 232 significantly increased PPARγ expression, when cells were treated with the individual fractions during the eight-day time course of differentiation. 

Correlation analysis of UCP-1/PPARγ expression revealed that 168, 232, 228 and 229 increased both UCP-1 and PPARγ expression. The fractions 139, 142, 143 and 155 resulted in overexpression of PPARγ, but low levels UCP-1 expression. The fractions 18, 37, 80 and 46 reduced both UCP-1 and PPARγ expression. However, reduction on both markers could also indicate toxicity and loss of cells (Figure 2C). 

Bright-field images of fully differentiated brown adipocytes at day 8 of differentiation that were treated with cyanobacterial fractions during differentiation are shown in Appendix A. Differences in brown adipocyte morphology were observed upon treatment with the fractions. We noticed that the adipocytes treated with fractions that showed an increased expression of UCP1 and PPARγ (fractions 228, 168, 229 and 232) appeared elongated compared to the adipocytes treated with fractions that reduced marker gene expression (fractions 18, 37, 46 and 80). None of the tested fractions induced obvious signs of toxicity resulting in cell loss.

### 2.4. Glucose Uptake in HepG2 Cells

The library of cyanobacterial fractions was screened for activity on glucose uptake using 2-(*N*-(7-Nitrobenz-2-oxa-1,3-diazol-4-yl)Amino)-2-Deoxyglucose (2-NBDG) in HepG2 cells. From the 263 analyzed fractions, five (1.9%) increased glucose uptake (MFI) >30%, while only one (0.4%) increased glucose uptake >50% (Figure 2D). Most promising fractions, characterized by consistent fluorescence values between replicates and no cytotoxicity, belong to four cyanobacterial strains, two from marine ecosystems (25 and 130) and two from freshwater ecosystems (48 and 77).

### 2.5. Toxicity Evaluation

Cytotoxicity was accessed in HepG2 cells following the glucose uptake assay using an MTT assay. Seven fractions revealed cytotoxicity higher than 30% (6, 7, 15, 32, 34, 119 and 149) and only one fraction revealed cytotoxicity greater than 50% (229). The remaining fractions did not show any cytotoxicity (Figure 2E). To determine the effect of cyanobacterial fractions on the viability of HepG2 cells, the sulforhodamine B (SrB) assay was performed following the anti-steatosis assay. The viability was reduced more than 30% by 50 of the fractions (19%) (3, 5, 12, 14, 17, 21, 22, 23, 24, 50, 52, 59, 69, 66, 67, 68, 69, 76, 79, 88, 99, 110, 116, 119, 121, 129, 135, 143, 145, 150, 160, 184, 187, 189, 190, 193, 196, 210, 214, 216, 217, 219, 222, 225, 226, 235, 246, 250, 259, 262), while 11 fractions (3.8%) (4, 57, 122, 130, 141, 142, 144, 154, 223, 229, 253) reduced the viability more than 50% (Figure 2F). General toxicity was evaluated in zebrafish larvae during the assessment of lipid homeostasis. Only one fraction (14) led to the death of all zebrafish larvae within 48 h of exposure, while the remaining fractions did not show any toxicity or malformations on the zebrafish larvae at the screening concentration of 10 µg mL^−1^.

### 2.6. Metabolite Profiling

The untargeted metabolite profiling of cyanobacterial strains was performed with an UPLC-QTOF MS platform. The different fraction types of cyanobacterial strains (e.g., fractions D) were individually analyzed. A principal component analysis (PCA) was used as the first step to identify metabolites from cyanobacterial fractions that substantially differ from the majority of the other metabolite profiles within the same fraction type. From 263 cyanobacteria fractions, 12 individual PCA´s were studied (one for each fraction type), as shown in Table 1. This analysis provided a number of markers ranging from 482 for fractions A to 1228 for fractions G, highlighting the chemical diversity of the fractions. Each of the markers was a single mass peak, characterized by its specific retention time and accurate mass. The first principal component (PC1) accounted from 11% to 27% of the total variance and the second from 9% to 15% for all analyzed fractions. Fraction I had a higher variability, while fractions E and H had the lowest variability. In the PCA´s, we searched for fractions that cluster differently compared to the majority of fractions of the same type, which should represent those fractions with the potential to produce different secondary metabolites. A summary of those fractions is given in Table 1, while all the corresponding PCA plots are shown in Appendix A.

As next step, we matched the obtained information regarding the (i) potential to produce different metabolites with (ii) bioactivity, and (iii) toxicity, in order to narrow the selection of promising fractions. Two examples are shown in Figure 3. The principal component analysis of the metabolite profiling of E fractions from all strains (Figure 3A) showed a central cluster with a relatively uniform distribution. The E fractions of LEGE03283 (#110), LEGE07175 (#130), LEGE07075 (#142) and LEGE06104 (#58) had a different distribution on the plot, but only the LEGE06104 fraction E (#58) showed anti-steatosis activity and no cytotoxicity, while the others did not have any anti-steatosis activity. The analysis of its chromatogram allowed the identification of several possible known compounds, but also contained a few peaks of possible unknown compounds with anti-steatosis activities (Appendix A). Figure 3B shows the principal component analysis of G fractions. The G fractions of the majority of the strains appeared together in one big cluster, except for LEGE07211 (#69), LEGE06174 (#122) and LEGE00246 (#180). From those three fractions, only fraction G of the LEGE00246 strain (#180) had strong bioactivity in the zebrafish fat metabolism assay and no toxicity. The analysis of the chromatogram revealed a potential for the isolation of new compounds. (Appendix A). From the total of 263 cyanobacterial fractions, a total of 66 (25% of 263 fractions) were identified with relevant bioactivities. The metabolite profiling and toxicity evaluation allowed the selection of the most promising 18 fractions (7%), which are summarized in Table 2. 

The identification of known and unknown compounds for one fraction of each bioactivity is represented in Figure 4, and the remaining chemical characterizations of promising fractions are shown in the Appendix A. By database searches in MarinLit [15], ChemSpider [16] and SciFinder [17], several known and unknown metabolites were identified, and the exact masses of the mass peaks compared to those available in the databases. For example, in Figure 4A, the search in databases resulted in the identification of eryloside T and a xanthin compound amongst several unidentified mass peaks in the fraction LEGE07173B with lipid reducing activity in zebrafish. In Figure 4B, the characterization of mass peaks identified leptosin F and a phaeophytin analogue in the fraction LEGE07167B with anti-steatosis activity. 

## 3. Discussion

The use of natural products as anti-obesity agents had been discussed in the literature [18,19], however, research in this field is still underexplored. While the striking potential of marine resources as producer of unique chemical structures is largely recognized, the exploration is often limited by the fact that the majority of species are not cultivable, by the lack of sustainable use of resources or by difficulties in chemical synthesis [20]. Phenotypic screening approaches regained the attention of many research groups and pharmaceutical companies [21]. These cell-based systems are efficient in detecting bioactivity of a compound but still poorly predict the *in vivo* characteristics. Small whole animal models were proposed to overcome the current limitation of the cell-based phenotypic assays, by adding a level of complexity to the models and incorporating a rudimentary safety test early in drug discovery [22]. In our study, we applied a blend of phenotypic cellular assays (lipid lowering and glucose uptake in HepG2 cells), target-based assays (adipocyte differentiation, and thermogenic gene expression) and the zebrafish Nile red fat metabolism assay [23] as a whole small animal model *in vivo* for lipid reduction. A total of 15 of the tested cyanobacteria fractions showed the capacity to reduce neutral lipids in zebrafish larvae *in vivo*. The characterization of the most promising fractions with lipid reducing activity led to the identification of the reported compounds eryloside T, pheophorbide A and phaeophytin A. Those compounds are minor peaks of the analyzed fractions and besides the literature reports them as bioactive, no obesity-related bioactivity is described. Phaeophyin A and pheophorbide A are two products of chlorophyll A degradation and were reported as effective anti-proliferative [24,25] and immunomodulatory [26,27,28] compounds. Those compounds seem to be widespread among the analyzed fractions, representing, in many of them, major peaks. Eryloside T, a minor peak in LEGE07173 B (#256), was previously isolated from the sponge *Erylus goffrilleri* and shown to have toxicity against Ehrlich carcinoma cells [29]. Lipid reducing activity of those compounds was not yet discovered. Other studies described bioactive compounds with lipid reducing potential in zebrafish larvae. *Talaromyces stipitatus*, a sponge-associated fungus, produces several anthraquinones and secalonic acid that show anti-obesity activity in the same animal model [10]. However, future studies will be necessary to study whether those known compounds or other present, unknown compounds in the fractions are responsible for the observed bioactivity. 

The liver is the main organ responsible for coordination of energy metabolism and lipid conversion. Non-alcoholic fatty liver disease (NAFLD) commonly appears as a result of excessive body fat gain [30]. As fat overloading works in a similar manner in primary hepatocytes and HepG2 cells, the latter represents a suitable model of hepatic steatosis [14]. A total of 50 fractions had shown anti-steatosis potential, however, many were cytotoxic, and finally, 17 fractions were selected with promising lipid lowering activity. Recent efforts have been made to discover novel molecules for the treatment of NAFLD. Siphonaxanthin, a marine carotenoid, isolated from green algae such as *Codium cylindricum Holmes*, showed a strong *in vitro* inhibitory effect of hepatic lipogenesis on the HepG2 cell line suppressing excessive lipid accumulation [31]. Another carotenoid, fucoxanthin, decreased lipid accumulation in FL83B hepatocytes [32]. Our metabolite profiling approach led to the identification of leptosin F, pheophorbide A, phaeophytin A and chlorophyll A in the promising fractions with anti-steatosis activities. Leptosin F, found as a minor peak, is an inhibitor of DNA topoisomerases I and II, important molecular targets of several potent anticancer agents, and cytotoxic effects in several tumor cell lines [33]. Chlorophyll A promotes similar activities as its degradation products, namely anti-proliferative [24,25] and immunomodulatory [26,27,28] activities. The pigment was, however, found as a minor peak. Hepatic lipid lowering of these compounds is not known yet. 

Hepatic glucose uptake and metabolism are important regulators of glycogen storage and *de novo* lipogenesis. To this end, four cyanobacterial fractions increased glucose uptake in HepG2 cells without any cytotoxicity. The characterization of those fractions identified, once again, chlorophyll A and its degradation products, as well as several carotenoids and terpenes. Carotenoids and terpenes are classes of compounds common in those fractions. Carotenoids are referenced for their diverse bioactivities such as anticancer, anti-inflammatory, cardioprotective, anti-obesity and anti-diabetic activities [34]. Terpenes are described for its health-related activities, such as cytotoxic, anti-microbial or anti-angiogenic [35,36]. These classes were, however, found in minority, when compared to unknown peaks. Similarly to our results, six polyoxygenated steroids, isolated from the marine sponge *Clathria gombawuiensis*, demonstrated a moderate increase in 2-NBDG uptake in 3T3-L1 adipocytes [37]. Other bioassays involving key macromolecules in DTM2 disease are more frequently applied, such as the activity of the tyrosine phosphatase 1B (PTP1B). PTP1B is a negative regulator of the insulin signalling pathway and considered a potential target to treat diabetes. Many compounds isolated from algae, like bromophenols, phlorotannins and sterols have shown strong inhibitory activity [38]. A bisabolane-related metabolite, isolated from the marine sponge *Axinyssa* sp., showed potent inhibitory effect of PTP1B [39]. 

The effect of the cyanobacterial fractions was also tested on brown adipocyte differentiation and thermogenic gene expression. For this purpose, a clonal brown preadipocyte cell line was used, selected for high differentiation capacity and robust expression of the mitochondrial uncoupling protein 1 (UCP-1), which mediates heat production through mitochondrial uncoupling. PPARγ is the key transcription factor promoting adipogenic differentiation and a well-established marker to assess adipogenesis [40]. UCP-1 gene expression was selected as indicator of thermogenesis (heat production), however, a confirmation by other methodologies will be necessary in the future to prove the positive effects on thermogenesis. Four fractions increased UCP-1 mRNA expression, and the metabolite profiling identified pheophorbide A, phaeophytin A, several terpenes and carotenoids, already identified in the previous fractions. The pattern seen before remains, and chlorophyll degradation products represent, in some cases, major peaks, but the other classes are minor. Several major peaks correspond to unidentified compounds that can represent novel structures. The low expression levels of UCP-1 and high expression of PPARγ upon treatment with four fractions were indicative for a shift of the treated cells towards a more energy-storing white adipocyte-like phenotype, while the reduction in both markers could indicate an inhibitory effect on adipogenesis in general. The previously mentioned fucoxanthin was described to induce UCP-1 expression in white adipose tissue [41]. Indeed, rats fed with the marine microalgae *Tisochrysis lutea* showed increased levels of both UCP-1 and PPARγ [42].

Metabolite profiling is an excellent tool to identify and select fractions/extracts for lead identification, as this technique is based on chemical profiling [43]. Due to the high amount of data acquired during metabolomic studies, statistical and multidimensional analysis are crucial for data mining and visualization. Principal component analysis (PCA) compares the variance between different samples. When applied to the cyanobacterial fractions, this technique allowed selecting the ones with a differentiating factor, representing different metabolites compared to the other samples. PCA, allied to metabolite profiling, has successfully resulted in the identification of several secondary metabolites, such as the 3-Alkyl Pyridine Alkaloids found in the sponge *Haliclona rosea* [44], or in the isolation of those metabolites, as N-Acyl-Taurine Geodiataurine, isolated from the polar sponge *Geodia macandrewi* [45]. In our study, the use of metabolite profiling and toxicity evaluation allowed to reduce the list of bioactive fractions to a handful of fractions with promising properties. Those properties have (i) relevant bioactivity towards diverse metabolic function, (ii) no toxicity *in vitro* and *in vivo*, (iii) the potential to produce novel compounds. After careful analysis of the compounds present in the cyanobacterial fractions, we expect that unknown compounds are causative for the tested obesity-related bioactivities. Although the activities for the identified compounds have never been studied in this context, they are only present as minor chromatographic peaks. Chlorophyll A and its degradation products could, however, be responsible for some of the reported activities. Future work is necessary in order to identify those compounds and to decipher the underlying molecular mechanism.

## 4. Materials and Methods

### 4.1. Growth of Cyanobacteria and Construction of Screening Library

Cyanobacterial strains used in this study were selected from Blue Biotechnology and Ecotoxicology Culture Collection [12] and are listed in Appendix A. All cyanobacteria were grown under a light/dark cycle of 14/10 h at 25 °C and a photon irradiance of approximately 30 µmol.m^−2^ s^−1^. Freshwater and estuarine strains were cultured in Z8 medium [46]. Marine strains were cultured in Z8 medium supplemented with 25 g L^−1^ NaCl and 20 μg L^−1^ vitamin B12. At the exponential growth phase, cells were harvested by centrifugation and freeze-dried. Lyophilized biomass was fractionated to obtain a screening library for the different bioassays. Two different approaches were used: vacuum liquid chromatography (VLC) fractionation or increasing polarity extraction.

#### 4.1.1. Vacuum Liquid Chromatography Fractionation

Lyophilized biomass was repeatedly extracted by percolation with a warm (<40 °C) mixture of dichloromethane:methanol (CH_2_Cl_2_:MeOH, 2:1 *v/v*) (VWR, Radnor, PA, USA). The resulting crude organic extract was separated using VLC with silica gel 60 (0.015–0.040 mm, Merck KGaA, Darmstadt, Germany) as stationary phase and a step-wise mobile phase gradient from 100% *n*-hexane (VWR) to 100% ethyl acetate (EtOAc) (VWR) and then to 100% methanol (MeOH) (VWR), yielding 10 fractions [47].

#### 4.1.2. Increasing Polarity Extraction

Lyophilized biomass was sequentially extracted with *n*-hexane, EtOAc and MeOH, with three extraction steps for each solvent. Briefly, each solvent was added to the biomass, and placed for 30 s in the ultrasonic bath, followed by 1 min in the vortex. Centrifugation at 4600 rpm for 15 min was performed and the supernatant collected. Three fractions were obtained, one for each solvent extraction. The fractions were dried and resuspended in dimethyl sulfoxide (DMSO) (VWR international, Radnor, PA, USA) at 10 mg mL^−1^ and stored at −20 °C.

### 4.2. Zebrafish Nile Red Fat Metabolism Assay

Lipid reducing activity was analyzed by the zebrafish Nile red fat metabolism assay [10,23]. Approval by an ethics committee was not necessary as chosen procedures are not considered animal experimentation according to the EC Directive 86/609/EEC for animal experiments. In brief, zebrafish embryos were raised from 1-day post fertilization (DPF) on in egg water (60 µg mL^−1^ marine sea salt) with 200 µM 1-phenyl-2-thiourea (PTU) to inhibit pigmentation. From 3 to 5 DPF, zebrafish larvae were exposed to cyanobacterial fractions at a final concentration of 10 µg mL^−1^ with daily renewal of water and fractions in a 48-well plate with a density of 6–8 larvae/well (*n* = 6–8). A solvent control (0.1% DMSO) and positive control (REV, resveratrol, 50 µM) were included in the assay. Neutral lipids were stained with 10 ng mL^−1^ Nile red overnight. The larvae were anesthetized with tricaine (MS-222, 0.03%) for 5 min before imaging on a fluorescence microscope (Olympus, BX-41, Hamburg, Germany). Fluorescence intensity was quantified in individual zebrafish larvae by ImageJ [48].

### 4.3. Cell Culture

HepG2 cells were purchased from American Type Culture Collection (ATCC) (Manassas, VA, USA) and cultured in Dulbecco Modified Eagle Medium (DMEM) (Gibco, Thermo Fisher Scientific, Waltham, MA, USA). Cells were grown in DMEM supplemented with 10% (*v/v*) fetal bovine serum (Biochrom, Berlin, Germany), 1% penicillin/streptomycin (100 IU mL^−1^ and 10 mg mL^−1^, respectively) (Biochrom) and 0.1% amphotericin (GE Healthcare, Little Chalfont, UK). HepG2 cells were incubated in a humidified atmosphere with 5% CO_2_, at 37 °C. A clonal cell line was derived from an SV40 large T immortalized brown preadipocytes, derived from the brown adipose tissue of a male C57Bl/6 mouse [49], based on high UCP-1 induction upon differentiation. Brown preadipocyte clones were cultured in normal growth medium (DMEM + GlutaMAXTM, 4.5 g L^−1^
d-glucose, pyruvate, 10% Fetal Bovine Serum (FBS) and 1% penicillin/streptomycin).

### 4.4. Anti-Steatosis Assay in HepG2 Cells and Sulforhodamine B (SRB) Assay

The anti-steatosis assay was adapted from [50] and [14]. Cells were seeded at 10^4^ cells/well in 96-well plates and adhered overnight. The medium was changed to DMEM supplemented with 62 μM sodium oleate (Sigma-Aldrich, St. Louis, MO, USA) and fractions were added at 10 μg mL^−1^. DMSO and 0.5% MeOH were used as solvent control, 62 μM sodium oleate as a negative control (maximum lipid accumulation) and resveratrol (REV) (Santa Cruz Biotechnology, Santa Cruz, CA, USA) as a positive control. After 6 h, cells were stained with 75 ng mL^−1^ Nile Red (Sigma-Aldrich) and 10 μg mL^−1^ Hoechst 33342 (HO-33342) (Sigma Aldrich) in Hanks Buffered Salt Solution (HBSS) (0.137 M NaCl, 5.4 mM KCl, 0.25 mM Na_2_HPO_4_, 0.44 mM KH_2_PO_4_, 1.3 mM CaCl_2_, 1.0 mM MgSO_4_, 4.2 mM NaHCO_3_, glucose free). After incubating at 37 °C for 10 min and in the absence of light, cells were washed twice with HBSS. Fluorescence was read in a Synergy HT Multi-detection microplate reader (Biotek, Bad Friedrichshall, Germany) at 485/572 nm excitation/emission for Nile red and 360/460 nm for HO-33342 [51]. 

Cytotoxicity of the fractions was tested on HepG2 cell line using the SRB (MP Biomedicals, LLC, Illkirch-Graffenstaden, France) colorimetric assay. Following the anti-steatosis assay, cells were fixed for 1 h at 4 °C, in the dark, adding 50% (*w/v*) ice-cold trichloroacetic acid (TCA) (Fisher Scientific, Loughborough, UK) to the culture medium. Cells were washed four times with deionized water and the plates air-dried. Then, 0.4% (*w/v*) SRB in 1% acetic was added to each well for 15 min, followed by five washes with 1% acetic acid. The plates were again air-dried and 10 mmol L^−1^, pH 10.5 Tris–HCl (VWR international, Gen-Apex) was added to each well. Absorbance was read at 492 nm with reference at 650 nm on a Synergy HT Multi-detection microplate reader (Biotek, Bad Friedrichshall, Germany).

### 4.5. Glucose Uptake Assay in HepG2 Cells and MTT Assay

To evaluate the potential for diabetes of the library fractions, the uptake of 2-(*N*-(7-Nitrobenz-2-oxa-1,3-diazol-4-yl)Amino)-2-Deoxyglucose (2-NBDG) (Life Technologies, Thermo Fischer Scientific) was measured as described in [52]. Briefly, HepG2 cells were seeded in 96-well plates at a concentration of 10^5^ cells/well. Then, 24 h after seeding, cells were starved on Hank’s Buffered Salt Solution (HBSS) for 16 h. Cells were then exposed to the fractions at 10 μg mL^−1^. Then, 0.5% DMSO was used as solvent control and Emodin (TargetMol, Boston, Massachusetts, USA) as positive control. After 2 h of incubation, 100 µM 2-NBDG was added to each well for 1 h. Cells were then washed twice with ice-cold HBSS and the fluorescence was measured at 485/535 nm (excitation/emission) at Fluoroskan Ascent CF (MTX Lab Systems, Bradenton, FL, USA). 

The MTT assay (3-(4,5-Dimethylthiazol-2-yl)-2,5-Diphenyltetrazolium Bromide) was used to assess the cytotoxicity of the fractions on the HepG2 cells following the glucose uptake screening. Cells were exposed to 0.2 mg mL^−1^ MTT and incubated at 37 °C for 2 h. The medium was removed and 100 μL DMSO added to each well. The absorbance was read at 570 nm on Synergy HT Multi-detection microplate reader.

### 4.6. Brown Adipocyte Differentiation

Differentiation of brown pre-adipocytes was induced by adding DMEM containing 10% FBS, 1% penicillin and streptomycin, 500 µM IBMX, 5 µM dexamethasone, 125 µM indomethacin, 1 nM Triiodothyronine (T3), 100 nM insulin and 1 µM rosiglitazone. After two days, the induction medium was replaced by freshly prepared differentiation medium (DMEM containing 10% FBS, 1% penicillin and streptomycin, 1 nM T3 and 100 nM insulin and until day 4, 1 µM rosiglitazone). The differentiation medium was changed every other day until the cells were fully differentiated at day 8. To study the effects of cyanobacterial fractions on brown adipocyte differentiation, preadipocytes were differentiated, as described above, in the presence of 10 µg mL^−1^ cyanobacterial fractions from the day 0 until day 6. Cell cultures were tested regularly negative for mycoplasm. Live cells were imaged with a Keyence (America Inc.; BZ-9000 BioRevo, Chicago, IL, USA) microscope, using a Nikon Plan- Apochromatic 20x/0.75 objective (Nikon, Japan). Bright field images were captured using ‘Multi-color image capturing software’ built in the BZ-9000 system. 

#### PPARγ and UCP-1 mRNA Expression by Real-Time PCR

For mRNA expression analyses, RNA from differentiated brown adipocytes was isolated using the QuickExtractTM RNA extraction kit (Epicentre Biotechnologies, Madison, WI, USA,), following the manufacturer’s instructions. Synthesis of cDNA was performed in a Thermo Cycler by using the high-Capacity cDNA reverse transcription kit (Applied Biosystem, Foster City, CA, USA), according to the manufactures protocol. Real-time PCR with SYBR green was performed using iTaqTM Universal SYBR^®^ Green Supermix (BIO-RAD, Hercules, CA, USA) in a CFX384 Touch Real-Time PCR Detection System (BIO-RAD). Relative mRNA expression was calculated after normalization by TATA-binding protein (Tbp) expression. Primer sequences are listed in the Appendix A. Differential expression levels were calculated via the ΔΔCt method [53].

### 4.7. Metabolite Profiling

The metabolic profiling of cyanobacterial extracts was performed using a Waters ACQUITY UPLC system (Waters, Milford, MA, USA), coupled to a Waters Synapt G1 mass spectrometer equipped with electrospray ionization (ESI) probe (Waters, Wilmslow, UK). The analytical column ACQUITY UPLC BEH C18 (2.1 × 100 mm, 1.7 µm) (Waters, Milford, MA, USA) was used for separation and was maintained at 60 °C. Mobile phase A was water with 0.1% formic acid and mobile phase B was acetonitrile with 0.1% formic acid. The flow was maintained at 0.45 mL min^−1^. A linear gradient was used from 65% to 100% B during the first 8 min, followed by a column clean up at 100% B for 0.5 min and reconditioning at the initial conditions for 1.5 min. The total chromatographic run time was 12 min. The sample manager was maintained at 10 °C. The samples were analyzed in positive ionization mode and the ionization source parameters were kept as follows: capillary voltage 3.5 kV; cone voltage 42 V; source temperature 125 °C; desolvation temperature 450 °C, at a flow rate of 700 L h^−1^ (N_2_); cone gas flow rate 50 L h^−1^. Data acquisition was carried out using MassLynx 4.1 and MarkerLynx XS was used for peak picking, alignment and identification of markers (Waters, Milford, MA, USA). Markers between 100 and 1500 Da were collected with an intensity threshold of 100 counts and retention time and mass windows of 0.10 min and 0.050 Da, respectively. The noise level was set to 5.00. Statistical analysis of the data was done using EZinfo 3.0 (Sartorius Stedim Biotech, Umea, Sweden) and SIMCA 15 (Sartorius Stedim Biotech).

## 5. Conclusions

This work demonstrates the potential of marine, estuarine and freshwater species of cyanobacteria, to produce secondary metabolites with relevant bioactivities towards several metabolic functions. The combination of screening assays with metabolite profiling and toxicity evaluation allowed the selection of a few, very promising fractions. Within such fractions, several known and unknown secondary metabolites were identified, however, major mass peaks corresponded to unknown compounds. In the future, the compounds responsible for the bioactivities will be isolated and structures elucidated, before exploring their role for the treatment of metabolic diseases.

## Figures and Tables

**Figure 1 marinedrugs-17-00280-f001:**
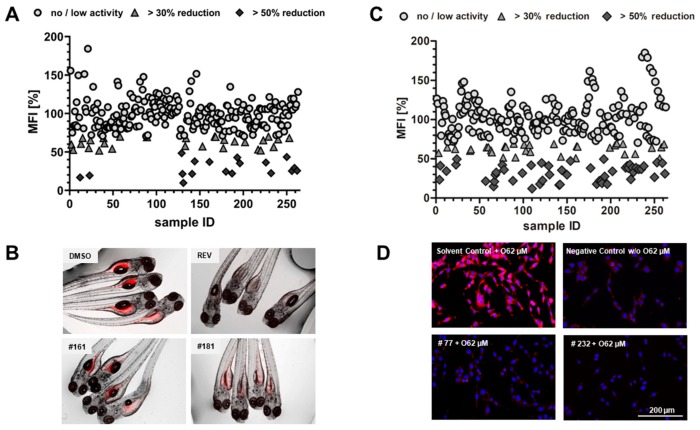
Bioactivity screening using the zebrafish Nile red fat metabolism assay and the anti-steatosis assay in HepG2 cells. (**A**) Data are presented as mean fluorescence intensity (MFI) relative to the solvent control. Zebrafish at 3 days post fertilization (DPF) were exposed for 48 h to 10 µg mL^−1^ cyanobacterial fractions and lipids around the yolk sac and intestine were stained with Nile red. (**B**) Representative images of zebrafish larvae (overlay of brightfield picture and red fluorescence channel). Solvent control, 0.1% dimethyl sulfoxide (DMSO); positive control, 50 µM REV and exposure to fraction #161 and #180. (**C**) Data are presented as MFI relative to solvent control (0.5% DMSO + O62 µM). HepG2 cells were exposed for 6 h to 62 µM sodium oleate (O62 µM) and 10 µg mL^−1^ cyanobacterial fractions. Nile red fluorescence stains neutral lipid reservoirs (red) and Hoechst 33342 the nucleus (blue). (**D**) Representative images of HepG2 cells (overlay of red and blue fluorescence channel). 0.5% DMSO + O62 µM; negative control, 0.5% DMSO; exposure to fraction #77 and #232.

**Figure 2 marinedrugs-17-00280-f002:**
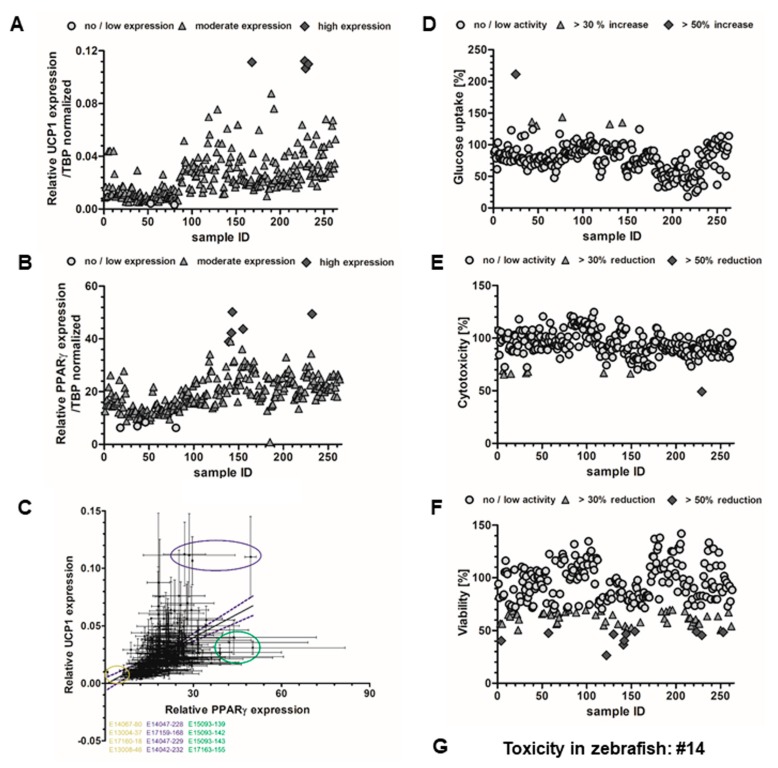
Analysis of mRNA expression of genes involved in (**A**) thermogenesis (uncoupling protein-1 (UCP-1)) and (**B**) adipocyte differentiation (PPARγ) by qPCR in brown adipocytes (*n* = 3). (**C**) Correlation between PPARγ and UCP-1 mRNA expression identifies three different groups (low UCP-1/low PPARγ; low UCP-1/high PPARγ; high UCP-1/high PPARγ). Values are shown as mean ± SEM. (**D**) Glucose uptake assay using 2-(*N*-(7-Nitrobenz-2-oxa-1,3-diazol-4-yl)Amino)-2-Deoxyglucose (2-NBDG) in HepG2 cells. Cells were exposed for 2 h to 10 µg mL^−1^ cyanobacterial fractions. An increase in fluorescence signal indicates higher uptake of the glucose analog 2-NBDG. Data are shown as mean fluorescence increase relative to the solvent control, 0.5% DMSO. (**E**) Cytotoxicity analysis by MTT from HepG2 cells following the glucose uptake assay. Data are presented relative to the solvent control, 0.5% DMSO. (**F**) Cytotoxicity of cyanobacterial fractions on HepG2 cells following the anti-steatosis screening assay. Data are presented in percentage relative to the solvent control (0.5% DMSO) + O62 µM. (**G**) The fraction 14 was the only to induce general toxicity in the zebrafish assay at 10 µg mL^−1^ (100% of mortality after 24/48 h of exposure).

**Figure 3 marinedrugs-17-00280-f003:**
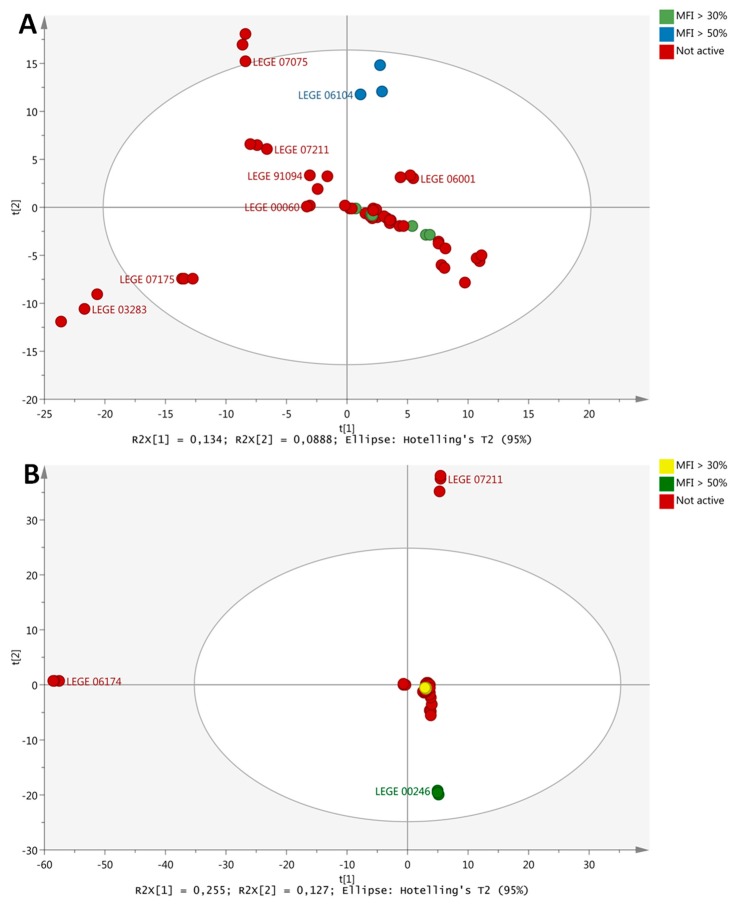
Matching of metabolite profiling with bioactivities for the selection of most promising cyanobacterial fractions. (**A**) PCA scores plot of E fractions colored according to anti-steatosis bioassay in HepG2 cells. Bioactivity is indicated as percentage MFI, mean fluorescence intensity. Strain names are indicated on the plot (e.g., LEGE07211). (**B**) PCA scores plot of G fractions colored according to activity in the zebrafish Nile red fat metabolism assay. The analysis was based on 1002 and 1228 collected markers on fraction E and G, respectively. The fractions were prepared in triplicate, and each replicate was run in triplicate.

**Figure 4 marinedrugs-17-00280-f004:**
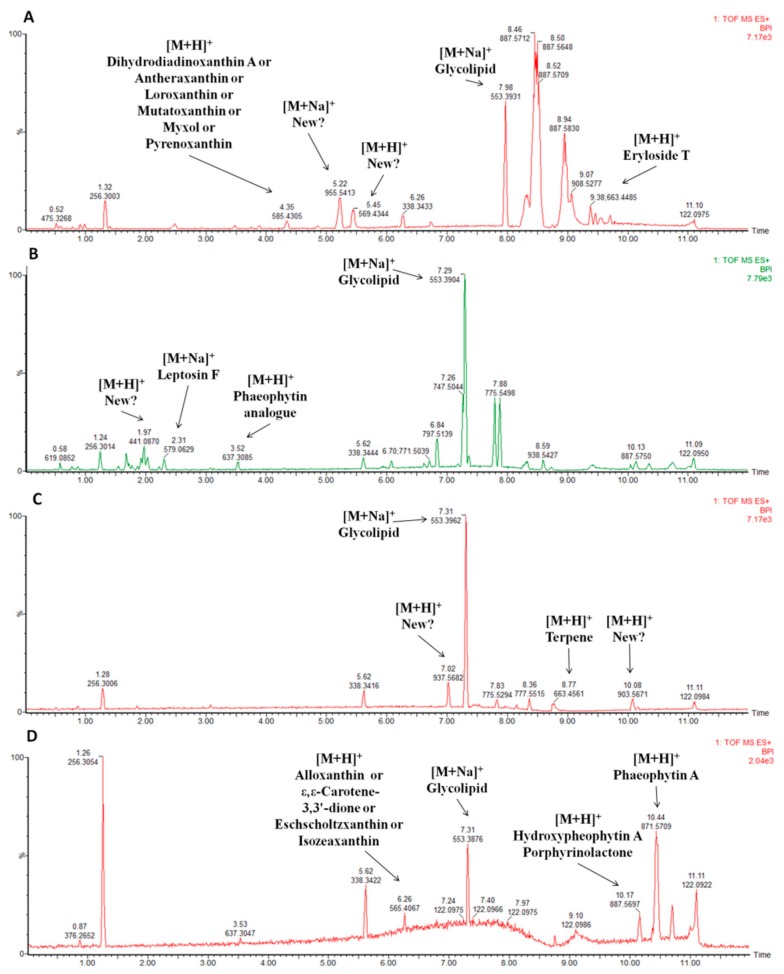
Base peak intensity chromatograms of selected cyanobacterial fractions with a respective identification of the compounds by database searches in MarinLit, ChemSpider and SciFinder. (**A**) Fraction LEGE 07173B (#256) selected based on zebrafish Nile red fat metabolism assay. (**B**) Fraction LEGE 07167B (#199), selected based on anti-steatosis bioassay. (**C**) Fraction LEGE 07172C (#77) selected based on glucose uptake bioassay. (**D**) Fraction LEGE 00247D (#168) selected based on PPARγ/UCP-1 expression levels. More information can be found in Appendix A.

**Table 1 marinedrugs-17-00280-t001:** Summary of principal component analysis (PCA) of the metabolite profiling of cyanobacterial fractions. Each marker was a single mass peak characterized by its specific retention time and accurate mass. Fractions that cluster differently were identified as those with the potential to produce different secondary metabolites, in comparison to fractions that clustered together, which were regarded to produce similar metabolites. IPE, increased polarity extraction; VLC, vacuum liquid chromatography; PC, principal component.

Fraction/Extract	Markers	Variance PC1 (%)	Variance PC2 (%)	Fractions with Potential to Produce Different Metabolites
**A (IPE)**	485	18	14	201, 216, 225, 228, 234, 240, 243, 249, 255, 258
**B (IPE)**	1131	19	9	199, 202, 205, 214, 217, 220, 226, 229, 235, 244, 253, 256
**C (IPE)**	1028	13	13	197, 221, 224, 248, 251, 254, 260
**A (VLC)**	815	15	11	1, 54, 75, 106
**B (VLC)**	628	19	11	20, 29, 46, 64, 139
**C (VLC)**	816	15	14	3, 12, 21, 77, 108
**D (VLC)**	943	16	10	19, 66, 88, 141
**E (VLC)**	1002	13	9	23, 40, 58, 67, 110, 130, 142, 160
**F (VLC)**	914	26	11	80, 121
**G (VLC)**	1228	26	13	69, 122, 180
**H (VLC)**	1178	11	10	92, 103, 123, 134, 154, 181
**I (VLC)**	1105	27	15	83, 164

**Table 2 marinedrugs-17-00280-t002:** Summary of most promising fractions with relevant bioactivities towards obesity, steatosis, diabetes or thermal energy release.

Bioactivity	Selected Fraction
Zebrafish—Lipid reducing	LEGE07175 H/#134
	LEGE00246 G/#180
	LEGE07172 A/#240
	LEGE07172 C/#242
	LEGE07173 B/#256
HepG2—Anti-steatosis	LEGE07084 D/#48
	LEGE03283 C/#108
	LEGE03283 D/#109
	LEGE07167 B/#199
	LEGE07160 B/#202
	LEGE06134 B/#220
HepG2—Glucose uptake	LEGE06001 G/#25
	LEGE06104 E/#58
	LEGE07212 C/#77
	LEGE07175 E/#130
Brown adipocytes—PPARγ and UCP-1 inducing activities	LEGE00247 D/#168
	LEGE06137 A/#228
	LEGE06097 B/#232

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
