# Peer review of "Identification of Cyanobacterial Strains with Potential for the Treatment of Obesity-Related Co-Morbidities by Bioactivity, Toxicity Evaluation and Metabolite Profiling"

_marinedrugs, 2019, doi:10.3390/md17050280_

Round 1
Reviewer 1 Report
I must confess I had a had time finding faults with this work. It presents a compelling case for the use of cyanobacterial-derived natural compounds for the study of obesity and related diseases.
The manuscript presents fairly good introduction and discussion sections, which require little to no improvement, in my opinion. Materials and Methods are also in par with the quality of the work.
My only "concern" relates to the authors relying exclusively in gene expression for the BAT cells assays. I could be useful to have some protein data, for example, a Western Blot of UCP1 and PPARg, for gene expression has some limits towards metabolic interpretation (for example, in the case of UCP1, a report by Nedergaard and Cannon clearly states this in its title: UCP1 mRNA does not produce heat; BBA, 2013; 1831(5):943-9.).
It could also be helpful to assess something about the mechanisms of glucose uptake in HepG2 cells (like GluT2 and 4 levels, for example), but I concede that this would fall out of the scope of this work and will fit more nicely in future works exploring the mechanisms behind the biological effects of these compounds.
Other than that, this manuscript is virtually ready for publication.
Author Response
I must confess I had a had time finding faults with this work. It presents a compelling case for the use of cyanobacterial-derived natural compounds for the study of obesity and related diseases.
The manuscript presents fairly good introduction and discussion sections, which require little to no improvement, in my opinion. Materials and Methods are also in par with the quality of the work.
My only "concern" relates to the authors relying exclusively in gene expression for the BAT cells assays. I could be useful to have some protein data, for example, a Western Blot of UCP1 and PPARg, for gene expression has some limits towards metabolic interpretation (for example, in the case of UCP1, a report by Nedergaard and Cannon clearly states this in its title: UCP1 mRNA does not produce heat; BBA, 2013; 1831(5):943-9.).
Answer: We fully agree with the reviewer that mRNA levels of PPARg and UCP-1 do not provide insights into the function of the respective proteins. However, in our screen we used the expression of PPARg and UCP-1 as markers for adipogenesis (PPARg) and brown adipocyte identity and thermogenesis (UCP-1). In our experience though, PPARg and UCP-1 protein levels closely follow the mRNA expression pattern. Moreover, the limited amount of cyanobacterial fractions available did not allow us to perform the differentiation in big enough cultures required for protein extraction and western blot validation. Thus, we would need to regrow all cyanobacterial strains in larger volumes and extract all fractions to assess the protein quantities of PPARg and UCP-1. While, we intend to do so in the near future, this is not possible in the time given for the revision of the current manuscript. We have added some clarification on this limitation in the text (results and discussion section).
It could also be helpful to assess something about the mechanisms of glucose uptake in HepG2 cells (like GluT2 and 4 levels, for example), but I concede that this would fall out of the scope of this work and will fit more nicely in future works exploring the mechanisms behind the biological effects of these compounds.
Answer: We agree. We intend to perform this work in the future.
Other than that, this manuscript is virtually ready for publication.
Reviewer 2 Report
Major comments:
The study is interesting, and I do hope that some of these new molecules will show in the future therapeutic anti-obesity activity. Nonetheless, it is yet to be determined which of the molecules that are detected in a bioactive cyanobacterial fraction are ones with the anti-obesity effect. The reader could be confused that all molecules within a bioactive fraction are potent. This should be clear already in the abstract – lines 29-32 – there’s no evidence that any of the mentioned isolated molecules have direct effects in your own hands, only the crude fractions. Please address it in the discussion already from the beginning and not only in the final paragraph.
The RNA expression in brown adipocytes may be insufficient to prove enhanced brown adipocyte activity. Please show it at a protein level, for example by UCP-1 immunofluorescence. No need to expose cells to all fractions, only to the fractions with positive and negative effects on UCP-1.
Figure 3 – please generate similar plots as examples for glucose uptake and for brown adipocyte activity, if possible.
Minor comments:
Figure 2A&B – you switched between qPCR results of PPARg and UCP-1. PPARg should be 2A and UCP-1 should be 2B according to the text in the results.
Figure 2D – Please correct to 50% and 30% instead of 150% and 130%, line 126 and in the figure itself. Please add the term “glucose uptake” on the Y-axis.
Figure 2E – Please add the term “cytotoxicity” on the Y-axis.
Figure 2F – Please add the term “viability” on the Y-axis.
The method of digging out anti-obesity fractions based on PCAs of metabolite profiling is creative and very interesting, however, it should be better explained. A reader may not know why it is done at the first place. For instance, mass-spec can be used to identify possible bioactive molecules in bioactive fractions, so why do you need to compare it to other fractions on a PCA plot. I know the answer, but please clarify it for the reader.
Author Response
The study is interesting, and I do hope that some of these new molecules will show in the future therapeutic anti-obesity activity. Nonetheless, it is yet to be determined which of the molecules that are detected in a bioactive cyanobacterial fraction are ones with the anti-obesity effect. The reader could be confused that all molecules within a bioactive fraction are potent. This should be clear already in the abstract – lines 29-32 – there’s no evidence that any of the mentioned isolated molecules have direct effects in your own hands, only the crude fractions. Please address it in the discussion already from the beginning and not only in the final paragraph.
Answer: We agree that we observe the activity in the fractions, and that we still have to find out whether identified compounds within the fractions are responsible for those activities or other unknown compounds also present in the fractions. A sentence was added to the abstract and in the beginning of the discussion in order to clarify this issue.
The RNA expression in brown adipocytes may be insufficient to prove enhanced brown adipocyte activity. Please show it at a protein level, for example by UCP-1 immunofluorescence. No need to expose cells to all fractions, only to the fractions with positive and negative effects on UCP-1.
Answer: We agree to the comment that UCP1 mRNA expression does not prove enhanced brown adipocyte activity. In fact, we are currently planning to test the positive and negative fractions using a Seahores Extracellular Flux Analyser to really test mitochondrial function and uncoupling. However, as we have mentioned above (answer to reviewer #1), the limited amount of fractions available to date, did not allow us to perform any assays exceeding the measurement of mRNA expression by qPCR, and in particular the differentiation in big enough cultures required for protein extraction and western blot validation. The preparation of fresh patches of cyanobacterial fractions is currently ongoing, but will still require several months to complete. However, we edited the text to make this limitation clear (results and discussion section).
Figure 3 – please generate similar plots as examples for glucose uptake and for brown adipocyte activity, if possible.
Answer: We assume that here Figure 2 is addressed. We have adapted the size and colour of the plot symbols across all panels. Regarding the brown adipocyte activity, we prefer to show the relative mRNA expression level, since this provides information of the general expression level and allows a better comparison to other studies.
Minor comments:
Figure 2A&B – you switched between qPCR results of PPARg and UCP-1. PPARg should be 2A and UCP-1 should be 2B according to the text in the results.
Answer: We have changed the text accordingly, and maintained the order of the figure.
Figure 2D – Please correct to 50% and 30% instead of 150% and 130%, line 126 and in the figure itself. Please add the term “glucose uptake” on the Y-axis.
Answer: Corrected as requested.
Figure 2E – Please add the term “cytotoxicity” on the Y-axis.
Answer: Corrected as requested.
Figure 2F – Please add the term “viability” on the Y-axis.
Answer: Corrected as requested.
The method of digging out anti-obesity fractions based on PCAs of metabolite profiling is creative and very interesting, however, it should be better explained. A reader may not know why it is done at the first place. For instance, mass-spec can be used to identify possible bioactive molecules in bioactive fractions, so why do you need to compare it to other fractions on a PCA plot. I know the answer, but please clarify it for the reader.
Answer: Thank you for your valuable comment. A brief explanation to the reason why PCA was used in our work was added to the discussion part.
Round 2
Reviewer 2 Report
No comments.